# Involvement of Nuclear Factor-κB in Inflammation and Neuronal Plasticity Associated with Post-Traumatic Stress Disorder

**DOI:** 10.3390/cells11132034

**Published:** 2022-06-27

**Authors:** Sudhiranjan Gupta, Rakeshwar S. Guleria

**Affiliations:** Biomarkers & Genetics Core, VISN 17 Center of Excellence for Research on Returning War Veterans, 4800 Memorial Drive (151C), Waco, TX 76711, USA; rakeshwar.guleria@va.gov

**Keywords:** PTSD, NF-κB, inflammation, neuronal plasticity

## Abstract

Post-traumatic stress disorder (PTSD) is a debilitating psychiatric condition which develops either due to stress or witnessing a traumatic situation. PTSD is characterized by acute and chronic stress response exhibit anxiety, fear, and an increased inflammatory etiology. Inflammation contributes a critical role in several parts of the brain that control fear and flashback cognatic function. It is known that impairment of the neurological circuit leads to the development of PTSD. Evidence has suggested that dysregulation of the sympathetic nervous system and hypothalamic-pituitary adrenal (HPA) axis and inflammatory responsiveness are pivotal and a greater risk in PTSD. NF-κB, a master regulator for inflammation, has been showed to modulate memory reconsolidation and synaptic plasticity; however, NF-κB’s association with PTSD remain elusive. In this review, we provide relevant findings regarding NF-κB activity in various components of brain and describe a potential mechanism linking PTSD using preclinical and clinical models. We envisage NF-κB signaling as a crucial mediator for inflammation, cognitive function, memory restoration and behavioral actions of stress and suggest that it could be used for therapeutic intervention in PTSD.

## 1. Introduction

Post-traumatic stress disorder (PTSD) is a severe distressing condition that develops in a subset of individuals after a major traumatic event and is associated with high morbidity and mortality [1,2]. In addition, PTSD is linked with deregulation of the sympathetic nervous system, hypothalamic-pituitary-axis (HPA) [3,4,5,6]. Recent evidence suggests that dysregulation of inflammation significantly contribute to development of PTSD [7,8]. However, the interplay between inflammation and neurocognitive pattern in PTSD is emerging. In the settings of therapeutic intervention, a deeper understanding of inflammatory pathology would be of interest for behavioral intervention.

In this review, we first describe pathophysiology of PTSD, then summarize evidence of dysregulation in the inflammatory circuit system in PTSD. We further describe the potential role of NF-κB, a master regulator of inflammation linking PTSD with the dysregulated inflammatory system by examining relevant findings from basic and clinical studies.

## 2. PTSD and the Pathophysiology

PTSD is a debilitating condition and one of the most pervasive psychiatric conditions resulted from traumatic stress. The term PTSD was first used in the third edition of the Diagnostic and Statistical Manual of Mental Disorders and was classified as an anxiety disorder [9]. In a subsequent revision in 1987, the diagnostic criteria were modified and emphasize the avoidance phenomena [10]. The inherent characteristics of PTSD are the alteration of mood, avoidance, increased arousal, hypervigilance, sleep disturbance, cognition deficit, flashbacks, reexperiencing traumatic events and persistence of intense fearful reactions [11]. PTSD is also a serious burden in deployed combat Veterans [12]. PTSD is frequently diagnosed in service members returning from war zones [13,14,15]. Typically, four essential features are examined during diagnosis and, they included traumatic events that caused injury, revisiting the angst periods with flashbacks, avoidance of past frightful memories and increased arousal. A significant increase of PTSD is observed in socially challenged people, young people, and first responders to trauma [16] and is associated with mood, anxiety and substance related issues [17]. Therefore, it is imperative to understand the alteration of brain physiology and the possible mechanism that caused the traumatic stress for therapeutic intervention.

The pathophysiology of PTSD necessitates the changes in neurotransmitter release and neurohormonal function in the brain. It is noted that an individual with PTSD showed a low level of glucocorticoid or cortisol [18] and high level of corticotrophin releasing hormone (CRH), a critical stress hormone [19]. The high level of CRH is observed in the cerebrospinal fluid indicated in PTSD patients indicated their association with the severity of the disorder [20,21]. In response to stress, hypothalamus releases CRH from paraventricular nucleus which activates norepinephrine neuro-modulatory system and triggers the behavioral “fight or flight” response from the SNS [22,23]. CRH acts on the pituitary gland, which in response secretes adrenocorticotropic hormone (ACTH) into the bloodstream. Once ACTH reaches the adrenal glands, it triggers the release of cortisol (in humans) or corticosterone (in rodents), which acts as anti-inflammatory hormones, and coordinates the physiological behavioral response to stress. Interestingly, single nucleotide polymorphism (SNP) in CRF1 receptor gene is reported to modulate stress susceptibility or resilience in severe depression [24,25]. Furthermore, SNP in the CRF1 receptor gene is reported in PTSD patients suggested that CRF signaling contributed to depression-related cognitive dysfunction pediatric PTSD subjects [26].

Essentially, two central physiological pathways are involved in PTSD; the hypothalamic pituitary adrenal (HPA) and sympathetic nervous system (SNS). The HPA-axis is a nodal point of major regulatory pathways controlling physiological and biochemical responses to stress. Alterations in glucocorticoid receptors (GRs) in HPA axis are implicated in the pathogenesis of PTSD [27]. Physiological stimuli or stress trigger HPA-axis leading to the secretion of glucocorticoid (GC) from adrenal cortex and mediating a negative feedback circuit during stressful situation [28]. Studies showed varied GR numbers, GR promoter methylation in PTSD conditions [29,30,31,32], however, an increased sensitivity of GR and inflammation are observed in PTSD cohorts [33,34].

The focal point of the central nervous system involved in fear is the amygdala. The almond-shaped mass in the cerebral hemisphere of brain structure primarily governs our ability to experience fear, emotions and extinguish memory of fearful stimuli [35,36]. The role of the amygdala in the pathophysiology of PTSD is still incompletely understood, however, recent functional magnetic resonance imaging (fMRI) technique has shown the ability to identify increased amygdala activity in PTSD following exposure to traumatic stress [37]. Patients with PTSD compared to non-PTSD subjects showed greater amygdala activation during exposure to an aversive stimulus and during extinction training [38,39]. Moreover, amygdala volume, a morphological feature in the structural plasticity of amygdala, is considered as critical factor in PTSD. The research conducted by Kuo et al. [40] using veterans with PTSD demonstrated a decreased amygdala volume in PTSD compared to the control group. Similar studies further confirmed the finding and claimed a relationship between smaller amygdala volume and stronger fear or stress response [41]. Recent studies showed that inflammatory response may be a contributor to enhanced amygdala activity. Neuroimaging studies demonstrated that increased amygdala activity to stress is associated with IL-6 secretion [42].

## 3. Neuronal Plasticity in PTSD

Neuronal plasticity may be defined as the ability to adapt nerve cells in the brain through changes in the growth, remapping or reorganizing of the neighborhood regions. It is also suggested that molecular reorganization of synapse may occur due to structural alteration of neuron in interneural communication process [43,44]. The brain areas that are involved in PTSD are prefrontal cortex, amygdala and hippocampus; tangled with long-term changes in the neurobiology of the brain. The neurohormonal machinery primarily act on brain areas to control PTSD symptoms like GR and norepinephrine. Therefore, a dysfunction in synaptic or neuronal plasticity with varied form is pivotal in PTSD symptoms. Below, we will discuss brain areas that are affected by plasticity.

### 3.1. Prefrontal Cortex and PTSD

The PFC participates in numerous cognitive functions including memory, processing, decision making, and behavior [45]. PFC is a heterogeneous structure and contains four functional parts based on dorsal/ventral and lateral/medial locations constituting dorsomedial prefrontal cortex (dmPFC), ventromedial prefrontal cortex (vmPFC), dorsolateral prefrontal cortex (dlPFC) and ventrolateral prefrontal cortex (vlPFC) regions. Both dmPFC and vmPFC regions are active in normal memory and fear processing, but, in PTSD cases, it is noted smaller activation particularly in war veterans [46]. Dysregulation of dlPFC and connection distribution in the cortical region resulted in hypoactivation in PTSD and considered as critical contributor in PTSD [47]. The repetitive transcranial magnetic stimulation (rTMS), a non-invasive stimulation using repetitive magnetic pulses in the brain tissue, used for various psychiatric diseases including PTSD, showed significant reduction of PTSD symptoms by targeting dlPFC [47]. Another veteran study reported that severity of PTSD is inversely correlated with anterior cingulate cortex volume [48]. However, the cause for hypoactivity of vmPFC is due to PTSD or a secondary effect is undetermined. Studies have also shown that miscommunication or disruption of vmPFC-amygdala was pivotal in the pathogenesis of PTSD and symptoms [49]. Also, Interestingly, studies demonstrated a reduction in structural integrity of white matter tracts in PTSD patients using diffusion tracer imaging [50]. The uncinate fasciculus white matter tracts, a long-range fiber tract that connects vmPFC to multiple subcortical, limbic regions including amygdala, has been proposed to play a role in language recognition. These facts together supports that dlPFC dysregulation in PTSD and targeting dlPFC structure and function possibly helps to ease PTSD symptoms. Therefore, structural and functional alteration in PFC in PTSD is critical and further research is warranted.

### 3.2. Amygdala and PTSD

The amygdala is an almond-shaped mass situated in the cerebral hemisphere controlling emotion. It is one of the key brain components necessary in the pathophysiology of PTSD. It is subdivided into 13 or more subnuclei; of which basolateral amygdala (BLA), central amygdala (CeA) and medial amygdala (MeA) were well documented [51]. The BLA contains excitatory neurons showing cortical-like profile whereas the CeA and MeA show striatal-like composition and are inhibitory neurons [50]. A greater engagement of amygdala activity is seen in PTSD patients [52,53,54]. An increased amygdala activity was conformed in fMRI study [55]. Similar activity was reported in Vietnam war veterans [46]; however, no activation was reported in conditions with PTSD [56]. Furthermore, the investigation of PTSD and amygdala volume are debatable; however, a trend showed smaller volume [57,58,59]. It is still unclear whether a smaller volume of amygdala is the outcome of PTSD or represented as a preexisting situation. It is therefore imperative to understand the role of amygdala activation or volumetry in the pathophysiology of PTSD, which is currently less understood.

### 3.3. Hippocampus

Hippocampus, a S-shaped structure and an integral part of limbic system plays a vital role in memory regulation. It is embedded in the temporal lobe of each cerebral cortex. It is a brain memory navigation module that is divided into three parts or cornu ammonis (CA) namely CA1, CA2 and CA3, respectively, constituting a long-term memory system [60]. Similar to amygdala, individuals with PTSD exhibit greater hippocampal activation during emotion processing [52]. A smaller hippocampal volume has been considered a risk factor to develop PTSD in women [61]. The integrity of hippocampus and its morphometric correlates are vital in degree or severity of PTSD [62,63]. In addition to CA, the hippocampus is divided two other subfields, the dentate gyrus (DG) and subiculum [64]. In the hippocampus, DG plays a critical role in learning, memory, pattern separation meaning ability to lessen the resemblance between two similar memories, contextual and spatial information [65]. Hippocampal changes specifically causing a reduction in DG volume was reported in a PTSD individual [66,67]. Furthermore, an association between smaller DG and major mood disorder was reported in postmortem MDD subjects, indicating its pivotal role in PTSD [68]. Together, this underscored that DG plays a role in maintaining PTSD symptoms, and future studies are required for deeper understanding of whether low volume of deficit of DG is the cause of PTSD.

Together, the above components of the brain underscore the importance of their function in the pathophysiology of PTSD. The morphological correlate, the size, or the volume appear to be critical in hyperarousal or negative behavioral events in PTSD.

## 4. NF-κB and PTSD

### 4.1. Inflammation and PTSD

The concept of inflammation in mental disorder or major depressive disorder appears to be unknown. However, there might be a link when an observation was published, dating back as an offshoot of interferon α treatment in chronic hepatitis C patients [69]. The study showed that 17% of patients developed psychiatric side effects after a year-long treatment with recombinant interferon α [69]. Subsequent studies have identified an association between immune deregulation and inflammatory response in several psychiatric disorders, including PTSD [70].

Inflammation is a predominant cellular process derived from cellular injury and it diseases with a goal to protect the injured site by releasing several inflammatory cells like neutrophils, monocytes or lymphocytes. These cells release several chemical mediators including enzymes and cytokines to alleviate the response [71]. Emerging evidence suggests that inflammatory condition can be activated by chronic psychological stress or mental illnesses, including PTSD, and may exert destructive effects on the brain. Importantly, inflammation is suggested to be a pivotal event of PTSD in combat veterans [72]. Findings from blood biomarkers along with genetic variants of several genes indicated that inflammation appears to be a key process in the pathology of PTSD. The association between PTSD and inflammation were observed primarily in blood inflammatory molecules including a wide spectrum of cytokines, and c-reactive protein (CRP), etc. Over the past several years, several studies have shown that an individual with PTSD elicits an enhanced level of blood cytokines such as IL-1β, IL-6, TNFα, etc. [73,74,75,76,77]. Interestingly, IL-6 appeared to be the most significant cytokine in PTSD individuals [7,78]. In addition to inflammatory blood biomarker identification, inflammatory genetic association or gene expression in PTSD are less studied. Bruenig et al. showed a polymorphism in TNFα gene promoter [TNFA -308 (rs1800629)] in PTSD in Vietnam war veterans [77]; however, the link between PTSD and TNFα polymorphism is not established. Another genome-wide association study of PTSD showed a link between inflammation and PTSD and single-nucleotide polymorphism (SNP) in the retinoid-related orphan receptor α (*RORA*) gene, rs8042149 [79]. Therefore, variants within inflammatory genes that encode cytokines, such as *TNF**α*, *IL-6* and *IL-1**β*, are warranted for further investigation in PTSD.

Another well-studied inflammatory marker is CRP. Studies have shown a positive correlation between CRP and PTSD where individuals with PTSD showed increased CRP levels compared to healthy controls [73,78,80,81]. However, mixed results were observed in different PTSD cohorts. No significant difference was observed between control and PTSD individual in a meta analyzed by Passos et al. [78,82]. Recent studies have reported that the severity of CRP and PTSD symptoms may not be considered as a specific biomarker for PTSD [83,84]. The etiology and pathophysiology of PTSD is complex and in this setting one possible reason for variability may lie in the genetic variant or SNPs. Ref. [85] have identified CRP SNP *rs1130864* to be associated with PTSD in a large urban sample with PTSD symptoms. However, they did not examine *CRP* SNPs and the severity of PTSD and CRP levels. The findings were further analyzed by Miller et al. who suggested that “*CRP* SNPs moderated the association between PTSD and CRP levels” [86].

The mechanism of inflammation and PTSD is unclear and complex [87]. One of the common mechanisms suggested altered HPA-axis and autonomous nervous system (ANS) [6,88]. The hypoactive HPA-axis and hyperactive of sympathetic nervous system (SNS) appear to a plausible cause for PTSD. Evidence indicated that dysregulation of the HPA-axis resulted in low cortisol level in the blood eventually developing an inflammatory state [89]. It is noted that cortisol is glucocorticoid and is an anti-inflammatory and immunosuppressive agent [90]. Glucocorticoid further autoregulates its level via negative feedback mechanism by binding to glucocorticoid receptors in the hypothalamus and pituitary. Furthermore, stress triggered the release of corticotrophin hormone (CRH) in the hypothalamus that stimulates SNS to produce catecholamines including norepinephrine, a prime cause for hyperarousal in PTSD [90]. Therefore, deregulation of the HPA-axis can influence chronic inflammation (low-grade) in PTSD.

Another mechanism postulated that the stress associated proinflammatory condition created a “sterile inflammation”, the inflammation in the absence of pathogenic response, triggered by the activation of pattern recognition receptors (PRR) which bind to damage (or danger)-associated molecular patterns (DAMP) [91,92]. DAMPs are endogenous molecules which increase in response to psychological stress and include a variety of different ligands, like heat-shock proteins, S100 proteins, high mobility group box 1, uric acids, and adenosine triphosphate, which have been shown to promote PTSD like condition in a stress model [7,92].

Overall, previous studies have suggested that there is a strong correlation between inflammation and PTSD. It is imperative to understand how peripheral inflammation influences neurocognitive function, including attention, processing and executive function in PTSD. Evidence has suggested that peripheral or central inflammation negatively affected cognitive response [93]; however, studies are limited in the setting of deeper understanding of inflammation and neurocognitive deficit in PTSD. Thus, we propose further research in the line of inflammatory response, clinical assessment and neurobiology of PTSD.

### 4.2. NF-κB and PTSD

The nuclear factor κB (NF-κB) is a ubiquitous transcription factor that regulates a wide array of cellular and molecular functions in diverse diseases. Since its discovery in 1986 by Sen and Baltimore [94] in immune cells (B-cells), it showed promise in many therapeutic interventions. In addition to its critical role in immune modulation and inflammation, NF-κB gained attention in the nervous system as it showed a pivotal role in neuronal plasticity, memory formation, synaptic processes, neurotransmission, and neuroprotection [95,96,97,98]. The central nervous system (CNS) hosts an array of different type of heterogenous cell population and the interplay between cell types and NF-κB activation is largely unknown. Specifically, NF-κB’s role in PTSD is largely unknown. A recent review by Dresselhaus and Meffert [99] illuminate the contribution of NF-κB in CNS.

The NF-κB family is composed of several members. In mammals, the NF-κB family comprises five members: NF-κB1 (p105/p50), NF-κB2 (p100/p52), RelA (p65), RelB, and c-Rel. NF-κB1 and NF-κB1 are synthesized as large polypeptides, separately, and p50 and p52 subunits are generated after the posttranslational cleavage. Under normal condition, NF-κB (mostly p50 and p65) is sequestered in the cytoplasm with its inhibitory partner, IκB proteins, primarily IκBα and IκBβ [100,101,102]. Upon stimulation, IκB is phosphorylated by the IκB kinase complex, the IKK, ubiquitinated, and consequently degraded by the 26S proteasome. The released NF-κB, translocated into nucleus, binds to a specific DNA element and activates several NF-κB dependent genes [101,103]. A wide range of factors including tumor necrosis factor α, interleukin-1, nerve growth factor, lipopolysaccharides, reactive oxygen species can activate NF-κB and depending on the type of stimulus the posttranslational modification occurs [104].

The role of NF-κB in the nervous system in not unknown. NF-κB modulates several physiological, pathological and development functions in chronic neurological disorders [105,106,107]. However, in a rodent model, it has been reported that NF-κB is activated in ischemia and neurodegenerative diseases [108,109,110]. Several studies have also documented increased levels of NF-κB activation in brain tissues of traumatic brain injury, focal ischemia and seizure [111,112,113]. Furthermore, NF-κB activation was observed in neuronal survival [114], and blockade of NF-κB showed loss of neuroprotection in neuron-specific deletion of NF-κB [115]. However, the role of NF-κB in PTSD remains elusive although few reports showed NF-κB in psychological rodent models. Kassed, C. et al. [116] for the first time reported lesser anxiety-like behavior using p50 null mice. Additionally, psychological rodent models showed that NF-κB played a critical role in anti-neurogenic and behavioral actions and suggested therapeutical targets for depression [117]. Another study using the predator scent stress rat model showed long-term NF-κB activation in the hippocampus and elicited PTSD-like behavioral coming up were reduced using NF-κB inhibitor (PDTC) and high dose of corticosterone [118]. The study is interesting as inhibition of NF-κB restored the behavioral process in a stress model mimicking PTSD. Moreover, increased NF-κB activation was observed in major depression patients with increased early life stress indicated a link between major depression, early life stress, and inflammation [119]. Future studies are warranted for the treatment of stress related clinical disorders.

In a clinical setting, there is no direct measurement of NF-κB activity in patients with PTSD. However, one study investigated NF-κB signaling using monocytes from PTSD individuals [120]. The authors examined specific transcription binding motifs in the promoter regions of differentially expressed genes in monocytes from PTSD patients, compared with healthy controls. The authors showed upregulation of NF-κB target genes in male and female patients with PTSD. In addition, authors observed downregulation of GR target genes in these patients. The study indicated that altered monocytes gene expression may influence inflammatory pathways in PTSD patients [120] as NF-κB is a master regulator for inflammation.

## 5. A predictive Scenario of NF-κB Activation in PTSD

It is reported that proinflammatory molecules are synthesized and released from several cell types in CNS including astrocytes, microglia and neurons along with immune cells like macrophages [121]. Stress triggers the synthesis of CRH in the paraventricular nucleus of the hypothalamus and stimulates further to SNS to produce catecholamines, including norepinephrine, which leads to the induction of a castade of proinflammatory cytokines, such as IL-1 and IL-6, via an NF-κB-dependent manner [122]. On the other side, GR, a 777 amino-acids nuclear receptor, modulates the transcription of many genes in the HPA-axis [123,124,125]. The GR binds to GC at its C-terminal ligand binding site, dimerizes and translocates into the nucleus, binds the GC responsive element (GRE) with the help of co-activators like GRIP1, and activates transcription [126,127]. Glucocorticoids are anti-inflammatory molecules and are known to bind to other transcription factors like NF-κB-p65 [128,129,130]. The mechanism of GR and NF-κB interaction has been extensively studied, and GR is shown to interact with NF-κB protein, (p65) via protein-protein interactions and impaired NF-κB transactivation [129,130,131]. Both GR and NF-κB activity is seen in several parts of brain regions during stress. During inflammation, it is observed that both GR and NF-κB are involved in glial cell activation, possibly due to presence of immune receptors that facilitate the synaptic plasticity [132]. Synaptic plasticity is altered during chronic stress [4,133,134]. During stress, proinflammatory and anti-inflammatory molecules are released by microglia and may alter synaptic plasticity [135]. The inflammatory condition may lead to compromised glial function by disrupting glutamate homeostasis [136]. It is also noted that both GR and NF-κB are present in neurons and thought to play a role in brain development and synaptic signaling [97,98,137,138]. It is reported that, in a chronic stress population, a reduced GR and increased NF-κB activity is observed [139,140]. The activation of NF-κB is linked with long-term synaptic plasticity, and long-term memory, and was found to be activated in long-term potentiation in rodent study [141,142]. Treating the mouse brain with NF-κB decoy prevented the long-term depression and reduced LTP (full form of LTP) [143]. Similarly, in p50, knockout model late-LTP is impaired and it is suggested that NF-κB-p50 is required for long-term spatial memory in the hippocampus [144]. Similarly, GCs are anti-inflammatory and immunosuppressive in nature and have been found to induce *I**κ**Bα* expression and blocked nuclear translocation of NF-κB in chronic rat stress model [145]. Therefore, the imbalance of GR and NF-κB interplay may be pivotal in a psychiatric related disease like PTSD and are suggested for the development of targeted strategy to combat inflammation.

Fear memory is one of the pivotal phases in acute and chronic PTSD, and research has been trying to reconsolidate fear memory to alleviate PTSD symptoms. In a rodent model it has been shown that NF-κB is critical in synaptic plasticity, animal behavior and long-term memory formation [146,147], and inhibition of NF-κB dampens the reconsolidation of memory [148]. Specifically, the subunits of p50 and p65 were involved in memory formation and synaptic activity [149,150]. Amygdala, a central region for fear conditioning in PTSD has shown that NF-κB activity is required in the basolateral amygdala for memory reconsolidation and inhibition of NF-κB disrupted the process [151]. The observation indicated that NF-κB might be a potential pharmacotherapy target for PTSD.

Microglia played a pivotal role in immune, synaptic plasticity and cognitive function by surveying neuronal environment, but various neurological disorders, including PTSD, disrupt the microglial function [152,153,154,155]. Microglial dysfunction regulates cognitive deficits associated with PTSD. A number of PTSD studies showed increased proinflammatory molecules [78], which were secreted mainly from activated microglia to combat the situation and returned to resting state [152,153]. However, in a chronic state, the microglia changed the morphology and become dysfunctional [153]. In an electric foot-shock PTSD mouse model, microglia are shown to be activated, and an increase in the number of microglia altered morphology, and reduced branches and dendritic spine density in the CA1 region of the hippocampus and Pfc in the Cx3cr1-GFP mice [153]. The changes are the reflection of synaptic dysregulation and memory impairments.

In summary, it may be suggested that, during PTSD, a dysfunctional HPA axis increased CRH to stimulate adrenocorticotropin secretion, leading to an imbalance in cortisol level that alters synaptic plasticity, activating NF-κB signaling and releasing proinflammatory cytokines.

## 6. Conclusions

Our review revealed that inflammation is prevalent in the PTSD population, and different parts of the brain serve as modules to orchestrate the neurological signal affecting the etiology of PTSD and reflecting the cognitive function. NF-κB signaling is critical in long-term memory formation, synaptic plasticity, proinflammatory cytokines surge, and behavioral function in the brain. Blockade of NF-κB mitigated the proinflammatory cytokines implicated in stress and depression, and could provide beneficial actions for the treatment of PTSD. Furthermore, there is convincing evidence supporting the role of NF-κB in synaptic plasticity, and memory reconsolidation in rodent models indicated its critical role in LTP induction and memory retention, particularly in amygdala. Collectively, the current data underscore a plausible role of NF-κB in modulating synaptic niche and coordinating the inflammatory response in acute and chronic stress situations in PTSD. The findings may provide a novel target for pharmacological intervention in an individual with PTSD. However, there are open questions for the future determination of NF-κB’s role in PTSD. First, a limited number of studies have examined NF-κB‘s contribution in human subjects and more PTSD cohorts are required to validate the observation. Second, NF-κB associated gene expression in the setting of cellular stress and inflammation during synaptic plasticity warrants further investigation and it could provide more insight at an organ-specific gene regulated network in the brain. Third, the selective regulation of NF-κB in learning and memory formation needs more in-depth information in the setting of cognitive behavior in PTSD. Fourth, the context of NF-κB signaling as anti-neurogenic and behavioral actions of stress requires further study.

## Data Availability

Not applicable.

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
