# Peer review of "Involvement of Nuclear Factor-κB in Inflammation and Neuronal Plasticity Associated with Post-Traumatic Stress Disorder"

_cells, 2022, doi:10.3390/cells11132034_

Round 1

Reviewer 1 Report

The work by Gupta and Guleria reviews the pathophysiology of PTSD, the neural plasticity in the disorder, and then focuses on the evidence relating the NF-kappa B transcription factor in both processes, inflammation, and Neuronal Plasticity in the Post-traumatic stress disorder.

The aim of the reviewers is to deepen the understanding of the inflammatory response associated with PTSD thru linking both to the reported alterations in the NF-kappa B pathway that is involved in inflammation and plasticity.

Main contribution of the review is the compilation of evidence that supports the hypothesis that inflammatory response observed in PTSD is linked to the cognitive and mnesic changes observed in pathology, via alterations in the NF-kappa B pathway produced by a dysfunctional HPA axis. NF-kappa B is a transcription factor that plays important roles in inflammation, the storage of memory information and neuronal plasticity, so the integration of this functions is welcome to interpret pathological and physiological functions.

The review is well structured, I find the comprehensiveness of the review topic covered, except in the points specifically addressed line by line. The references are appropriate, and I think the review is of great relevance and actuality, linking behavioral and cognitive traits to inflammation and the transcription factor NF-kappa B.

line by line

Lines 90 and 91. “We will discuss below of brain areas that are affected by plasticity.” Many areas of the brain, if not all undergo some type of plasticity, and the author focus on prefrontal cortex, amygdala, and the hippocampus. I think the reason for election of these areas to be described should better defined, in terms of alterations observed in PTSD or the criteria used.

Lines 124, 125. “It is a brain memory navigation module is divided into three parts or cornu ammonis (CA) such as CA1, CA2 and CA2, respectively, constituting a long-term memory system [56].”  The second CA2 in this paragraph probably refers to CA3. I think the dentate gyrus should be generally mentioned, and there is evidence that the NF-kappa B pathway is altered in the dentate gyrus in response to stress, like in Cohen et al., 2011. This important input of the hippocampus (DG) should be at least mentioned. I also suggest that the DG may be of particular interest to mention due to its role in Pattern separation, a process resolving interference in encoding and retrieving similar experiences.

157 Bruenig D in capital letters no clear reason.

172 Michopoulos et al [78] underlined no clear reason.

The evidence related to inflammation and activation of NF-kapa B and GR is well covered in the sections NF-κB and PTSD and A predictive scenario of NF-kB Activation in PTSD. But I think that the review will gain directly addressing the complexity of NF-κB signaling in normal physiology and pathology to the lens of cell-type specific responses, citing Dresselhaus 2019.

Author Response

We thank the reviewers for their constructive comments. As per their recommendations, we have addressed all their concerns and it is our understanding that the overall impact of the review has increased significantly. We have incorporated our changes as per reviewers’ suggestion in track change version. Please find below our responses:

Reviewer 1

The reviewer 1 mentioned that the review is well structured, and the comprehensiveness of the review topic is covered. The references are appropriate, and the review is of great relevance and actuality, linking behavioral and cognitive traits to inflammation and the transcription factor NF-kappa B.

Response: We appreciate reviewer’s comment.

Query: Lines 90 and 91. “We will discuss below of brain areas that are affected by plasticity.” Many areas of the brain, if not all undergo some type of plasticity, and the author focus on prefrontal cortex, amygdala, and the hippocampus. I think the reason for election of these areas to be described should better defined, in terms of alterations observed in PTSD or the criteria used.

Response: Thank you for your suggestion. In the revised review, we have discussed the relevance of brain areas.

Query: Lines 124, 125. “It is a brain memory navigation module is divided into three parts or cornu ammonis (CA) such as CA1, CA2 and CA2, respectively, constituting a long-term memory system [56].”  The second CA2 in this paragraph probably refers to CA3.

Response: The reviewer is correct. It was a typo and we have corrected it in the revised version of the review.

Query: I think the dentate gyrus should be generally mentioned, and there is evidence that the NF-kappa B pathway is altered in the dentate gyrus in response to stress, like in Cohen et al., 2011. This important input of the hippocampus (DG) should be at least mentioned. I also suggest that the DG may be of particular interest to mention due to its role in Pattern separation, a process resolving interference in encoding and retrieving similar experiences.

Response: We appreciate reviewer’s suggestion. In the revised version of the review, we have discussed dentate gyrus and PTSD.

Query: 157 Bruenig D in capital letters no clear reason.

Response: It was an error during formatting  and  is now corrected in the revised version of the review.

Query: 172 Michopoulos et al [78] underlined no clear reason.

Response: It was an error during formatting and was corrected in the revised version of the review.

Query: The evidence related to inflammation and activation of NF-kapa B and GR is well covered in the sections NF-κB and PTSD and A predictive scenario of NF-kB Activation in PTSD. But I think that the review will gain directly addressing the complexity of NF-κB signaling in normal physiology and pathology to the lens of cell-type specific responses, citing Dresselhaus 2019.

Response: In the revised version of the review, we have discussed Dresselhaus’s review.

Reviewer 2 Report

In the manuscript Cells-1775172, authors have described the pathophysiology of post-traumatic stress disorder and discussed the neuronal plasticity in post-traumatic stress disorder. Authors have reviewed basic science studies as well as clinical reports to discuss the potential role of nuclear factor-kappa B in inflammation associated with post-traumatic stress disorder.

Post-traumatic stress disorder is developed after a major traumatic incident and is associated with high morbidity. It is causally link with the deregulated sympathetic nervous system, hypothalamic-pituitary-axis.

Further, the dysregulated inflammatory response is suggested to contribute to the development of post-traumatic stress disorder. Considering the growing evidence for the interplay between inflammation and neurocognitive pattern in post-traumatic stress disorder, in depth understanding of inflammatory pathology of PTSD is of interest for the therapeutic development. In view of the above points, this manuscript by Gupta & Guleria is a timely review article.

It is comprehensive review and reflects great efforts of authors. It covers all the relevant studies as cites more than 140 references.

Major-

1.    This manuscript has no “Abstract”. Please consider drafting an abstract as per the journal’s guidelines.

Minor-

1.    Title “Post-traumatic stress disorder: Involvement of Nuclear factor-2 kB in Inflammation and Neuronal Plasticity” can be revised. Both segments don’t seem coherent on reading. For instance, “Involvement of Nuclear factor-2 kB in Inflammation and Neuronal Plasticity associated with post-traumatic stress disorder” could be closer to the review content organization. On a similar note, current title emphasizes on NF-kB, but NF-kB is introduced on 4th main heading.

2.    In such a comprehensive review, for the sake of readership, I would recommend having a few schematic diagrams to summarize the important messages or highlights.

Author Response

We thank the reviewers for their constructive comments. As per their recommendations, we have addressed all their concerns and it is our understanding that the overall impact of the review has increased significantly. We have incorporated our changes as per reviewers’ suggestion in track change version. Please find below our responses:

Reviewer 2

The reviewer 2 stated that “It is comprehensive review and reflects great efforts of authors. It covers all the relevant studies as cites more than 140 references”.

Response: We appreciate reviewer’s comment.

Major-

Query: This manuscript has no “Abstract”. Please consider drafting an abstract as per the journal’s guidelines.

Response: The abstract was uploaded during submission. We apologized that the reviewer did not receive it at the other end.  We will make sure that it is received by the reviewer during our second submission (revised version). 

Minor-

Query: Title “Post-traumatic stress disorder: Involvement of Nuclear factor-2 kB in Inflammation and Neuronal Plasticity” can be revised. Both segments don’t seem coherent on reading. For instance, “Involvement of Nuclear factor-2 kB in Inflammation and Neuronal Plasticity associated with post-traumatic stress disorder” could be closer to the review content organization. On a similar note, current title emphasizes on NF-kB, but NF-kB is introduced on 4th main heading.

Response: As per the suggestion, we have modified the title.

Query:  In such a comprehensive review, for the sake of readership, I would recommend having a few schematic diagrams to summarize the important messages or highlights.

Response: We have schematic diagram in the review.